

**Measurement Report: Size-resolved secondary organic aerosol formation modulated by aerosol water uptake in wintertime haze**

Jing Duan[1], Ru-Jin Huang[1,2], Ying Wang[1], Wei Xu[3], Haobin Zhong[4], Chunshui Lin[1], Wei Huang[1], Yifang Gu[1], Jurgita Ovadnevaite[5], Darius Ceburnis[5], Colin O'Dowd[5]

[1]State Key Laboratory of Loess and Quaternary Geology (SKLLQG), Center for Excellence in Quaternary Science and Global Change, Institute of Earth Environment, Chinese Academy of Sciences, Xi'an 710061, China

[2]University of Chinese Academy of Sciences, Beijing 100049, China

[3]Center for Excellence in Regional Atmospheric Environment, Institute of Urban Environment, Chinese Academy of Sciences, Xiamen, China

[4]School of Advanced Materials Engineering, Jiaxing Nanhu University, Jiaxing, 314001, China

[5]School of Physics and Centre for Climate and Air Pollution Studies, Ryan Institute, University of Galway, University Road, Galway, H91CF50, Ireland

**Correspondence**: Ru-Jin Huang (rujin.huang@ieecas.cn)

**Abstract**

This study investigated the potential effects of inorganics changes on aerosol water uptake and thus secondary organic aerosol (SOA) formation in wintertime haze, based on the size-resolved measurements of non-refractory fine particulate matter (NR-PM$_{2.5}$) in Xi'an, Northwest China. The composition of inorganic aerosol showed significant changes in winter 2018-2019 compared to winter 2013-2014, shifting from a sulfate-rich to a nitrate-rich profile. In particular, the fraction of sulfate and chloride decreased but nitrate increased in the entire size range, while ammonium mainly increased at larger particle sizes. These changes thus resulted in size-dependent evolution in water uptake. Increased water uptake was observed in most cases mainly associated with enhanced contributions of both nitrate and ammonium, with the highest increase ratio reaching 5-35% at larger particle sizes and higher relative-humidity (RH). The non-negligible influence of chloride on aerosol water uptake was also emphasized. The random forest analysis coupled with a Shapley additive explanation algorithm (SHAP) further showed enhanced relative importance of aerosol water in impacting SOA formation. Aerosol water contributed to the SOA formation in most cases in winter 2018-2019, and the SHAP value increased as aerosol water increased, especially at larger particle sizes. This implies the majority of enhanced aerosol water uptake at larger particle sizes and high RH might facilitate the efficient aqueous-phase SOA formation. This study highlights the key role of aerosol water as a medium to link inorganics and organics in their multiphase processes. As challenges to further improve China's air quality remain and SOA plays an increasing role in haze pollution, these results provide an insight into the size-resolved evolution characteristics and offer a guidance





for future control.

## 1. Introduction

Particulate matter with a diameter of less than 2.5 μm (PM$_{2.5}$) has become a major concern in air pollution, which is associated with adverse health effects and regional climate change (Huang et al., 2014; Cai et al., 2017; Burnett et al., 2018; An et al., 2019; Wu et al., 2022). During the past two decades, particulate pollution in China has attracted widespread attention, and many studies have been carried out to elucidate the aerosol characteristics and formation mechanisms in haze pollution (Zhang et al., 2015; Huang et al., 2019; Zhong et al., 2020; Zheng et al., 2021; Gu et al., 2023). Aerosol characteristics such as mixing state, hygroscopicity, and acidity are strongly influenced by particle size. The size distribution of aerosol components provides essential information to elucidate the formation and transformation processes in the atmosphere, e.g., the formation and growth of new particles, or photochemical and aqueous-phase reactions (Craig et al., 2018; Kuang et al., 2020; He et al., 2021; Xu et al., 2021; Cai et al., 2022). Therefore, analyzing the characteristics of size-segregated aerosol is instrumental in comprehending their formation mechanisms in the atmosphere.

In recent years, China has implemented a series of control measures to improve air quality, including the introduction of desulfurization equipment, stricter emission standards for coal-fired power plants and industries, limitations on the number of on-road vehicles, and the replacement of coal combustion with natural gas (Fontes et al., 2017; Zheng et al., 2018; Zhang et al., 2020). As a result of these measures, the annual average concentration of PM$_{2.5}$ in China has largely decreased, and air quality has improved (Zhang et al., 2019, Wang et al., 2020; Li et al., 2021). However, further improvement of China's air quality remains a challenge and requires more studies (Xu et al., 2019; Sun et al., 2020; Wang et al., 2022). A number of recent studies have shown changes in aerosol composition and properties since the state's two-phase program to clean China's air from 2013-2020, and many studies highlight the increasing contribution of secondary aerosols and the growing role of nitrate over sulfate in haze pollution (Xu et al., 2019; Li et al., 2021; Duan et al., 2022; Zhong et al., 2022; Lin et al., 2023). This is consistent with the faster decline in SO$_2$ concentration than in NO$_2$ concentration in recent years (Chu et al., 2020). Nitrate and sulfate are two main hygroscopic substances that absorb water and provide an aqueous environment on particles, facilitating multiphase chemistry (Herrmann et al., 2015; Wu et al., 2018). For instance, condensed water enables the partitioning of water-soluble, polar organic precursors into the aqueous phase, promoting the formation of secondary organic aerosol (SOA) (Wu et al., 2018, Lv et al., 2023). The water uptake capacity of aerosols is closely related to their chemical composition, mixing state and size distribution (Liu et al., 2019; Kim et al., 2020). The transformation of sulfate and nitrate roles in haze pollution will inevitably induce changes in aerosol water uptake capacity. However, the size-segregated changes of aerosol composition and water uptake capacity, as a consequence of air pollution



control in recent years, are not well understood, and thus their impact on the SOA formation remains unclear (Xu et al., 2019; Lv et al., 2023).

In this study, we analyzed the size-resolved NR-PM$_{2.5}$ composition and OA sources in the winter 2018-2019, and compared to those in the winter 2013-2014 in Xi'an, Northwest China. Particularly, we examined the size-resolved changes in inorganic species, and discussed the size-segregated evolution of aerosol water uptake induced by the changes in inorganics. Moreover, we investigated the potential influence of aerosol water uptake changes on SOA

formation processes in wintertime haze.

## 2. Experimental methods

### 2.1 Sampling and instrumentation

The campaigns in the winters of 2013-2014 and 2018-2019 were conducted at the campus of the Institute of Earth Environment, Chinese Academy of Sciences (34°23′N, 108°89′E) in

downtown Xi'an, surrounded by traffic, residential, and commercial areas. Detailed descriptions of the samplings are provided in Elser et al. (2016) and Duan et al. (2022), respectively. For comparative analysis, data from December 13[th], 2013 to January 6[th], 2014 and from December 23[rd], 2018 to January 20[th], 2019 were selected for this study, considering the similar time duration and both including clean and haze periods. The size-resolved NR-PM$_{2.5}$

composition was measured using an HR-ToF-AMS (Aerodyne Research Inc.) in the winter of 2013-2014, and using an SP-LToF-AMS (Aerodyne Research Inc.) in the winter of 2018-2019, respectively. Detailed information on instrument operation and calibration can also be found in Elser et al. (2016) and Duan et al. (2022).

### 2.2 Data analysis

The SQUIRREL (version 1.61D) and PIKA (1.21D) software, coded in Igor Pro 6.37 (WaveMetrics), were used to analyze the size-resolved data from HR-ToF-AMS in 2013-2014 and from SP-LToF-AMS in 2018-2019. Standard RIEs of 1.4, 1.1 and 1.3 were used for organics, nitrate and chloride, respectively. Experimentally determined RIEs of 1.48, and 3.37 in 2013-2014, and 1.30 and 3.70 in 2018-2019, were used for sulfate and ammonium, respectively.

Additionally, incomplete detection due to particle bounce was calibrated based on the composition-dependent collection efficiency (CDCE) both in 2013-2014 and 2018-2019 (Middlebrook et al., 2012). The data and error matrices were preprocessed to obtain size-resolved NR-PM$_{2.5}$ composition (Org, SO$_4^{2-}$, NO$_3^-$, NH$_4^+$, and Cl$^-$) as well as OA mass spectra for m/z 12-120.

### 2.3 Size-resolved OA source apportionment

Positive matrix factorization (PMF, Paatero and Tapper, 1994) and multilinear engine (ME-2) in Igor Pro (Paatero, 1999) were used to perform the source apportionment for the size-resolved





OA mass spectra. We first binned the OA mass spectra in the range of 80-2500 nm into 10 size
ranges (Table S1). As the mass spectra of OA in the range of 80-113 nm and 1772-2500 nm in
2013-2014 contained many negative m/z values, we primarily performed the OA source
apportionment and results discussions for the mass spectra of the remaining 8 size bins in the
range of 113-1772 nm in both 2013-2014 and 2018-2019. The error matrices of size-resolved
OA mass spectra were created using Eq (1) according to the methods in Ulbrich et al. (2012)
and Polissar et al. (1998).


$$U_{ij}= \begin{cases} \dfrac{5}{6}LOD_i & if \ C_j \leq LOD_i \\[2ex] \sqrt{u_i{}^2 \times C_j{}^2 + LOD_i{}^2} & if \ C_j > LOD_i \end{cases} \quad (1)$$

where $U_{ij}$ refers to the uncertainty for the $i^{th}$ species in the $j^{th}$ row, and $C_j$ refers to the measured
mass loading. The three times standard deviations (3s) of the 5% lowest OA mass loading
during the measurement periods were applied as the limits of detection (LOD). The relative
uncertainties (u) of 20% were used for the particles, according to the transmission efficiency of
the aerodynamic lens. Before PMF analysis, the ions with signal-to-noise (S/N) < 0.2 were
removed and those with 0.2 < S/N < 3 were down-weighted (Xu et al., 2021).

Unconstrained PMF analysis with varied factor numbers was performed for OA mass spectra
in each size bin during winter campaigns of 2013-2014 and 2018-2019. The optimal factor
number was selected based on the principle that decreasing the number of factors causes mixing
between different sources, and increasing the number of factors leads to factor splitting and the
occurrence of non-meaningful factors. The results were evaluated by comparing the mass
spectral profiles with previous studies and the correlations with tracer species. As shown in Fig.
S1 and S2, two OA sources including primary organic aerosol (POA) and SOA were resolved
in each size bin in 2013-2014, and three OA sources including fossil-fuel-related OA (FFOA),
biomass burning OA (BBOA) and SOA were resolved in each size bin in 2018-2019. The
evolution and comparison of size-resolved OA sources will be detailed in section 3.3.

### 2.4 Calculation of aerosol liquid water content (ALWC) and mass growth factor for inorganic aerosols ($G_{mi}$)

The thermodynamic model ISORROPIA-II was used to calculate the ALWC. We ran the model
in "reverse" mode, in which inorganic aerosol components ($NH_4^+$, $SO_4^{2-}$, $NO_3^-$, $Cl^-$) combined
with ambient temperature (T) and relative humidity (RH) were input as parameters (Fountoukis
and Nenes, 2007). In addition, the simulation was run in "metastable" mode and all components
were assumed to be deliquescent. The thermodynamic equilibrium and phase state of those
inorganic species were then simulated and the ALWC ($\mu g \ m^{-3}$) was calculated. The mass growth
factor for inorganic aerosols ($G_{mi}$) was further calculated using the following equation:



$$G_{mi} = \frac{ALWC + m_i}{m_i}$$

Where $m_i$ is the dry mass concentration of inorganic aerosol (μg m$^{-3}$) (Song et al., 2019).

### 2.5 Random forest modelling

A decision tree-based random forest model was developed to understand the observed trend of
SOA. We analyzed the relationship between SOA and its influencing parameters, such as RH,
T, Ox (O$_3$+NO$_2$), solar radiation (SR), and aerosol liquid water. In the RF model, the whole
dataset was randomly divided into a training dataset and a testing dataset. The training dataset
contained 70% of the entire data, and was used to build the model. The test data contained the
remaining 30% of the entire data, and was used to validate the model performance. To grow a
tree, the number of independent/explanatory variables was set to 3, the minimum nod-size was
set to 5, and the number of trees in the forest was set to 300, according to Grange et al. (2018)
and Wang et al. (2022). Moreover, the SHapley Additive explanation (SHAP) algorithm was
applied to evaluate the importance of the variables in predicting SOA, which can help produce
an interpretable model and compute the importance of variables with physical significance
(Lundberg et al., 2020).

## 3. Results and discussion

### 3.1 Size-resolved changes in NR-PM$_{2.5}$ composition

The average size distribution of NR-PM$_{2.5}$ composition in winter Xi'an, as well as the relative
changes between 2018-2019 and 2013-2014 are shown in Fig. 1. The peak diameter of total
NR-PM$_{2.5}$ in 2018-2019 (631-890 nm) was smaller than that in 2013-2014 (890-1256 nm). The
mass fraction of total secondary inorganic species (SIA, including sulfate, nitrate, and
ammonium) in total NR-PM$_{2.5}$ increased from 28% to 58% as the particle size increased from
112 nm to 1772 nm in 2013-2014. Specifically, the mass fraction of sulfate displayed the largest
increase from 8% to 33%, followed by nitrate from 8% to 15%, while ammonium maintained
a relatively stable contribution of 10% across the entire size range. In contrast, in 2018-2019
the mass fraction of SIA in total NR-PM$_{2.5}$ increased from 25% to 51% as the particle size
increased from 112 nm to 1772 nm. Unlike in 2013-2014, the mass fraction of nitrate showed
the most significant contribution from 12% to 25%, followed by sulfate from 6% to 16%, and
ammonium from 8% to 11% in 2018-2019.

The mass concentrations of all NR-PM$_{2.5}$ species decreased significantly in 2018-2019
compared to 2013-2014 for the entire size range, and the reduction ratio decreased as particle
size increased from 112 nm to 1772 nm (Fig. 1e). However, the contribution of each species
showed different variations in 2018-2019 compared to 2013-2014. In particular, the
contributions of sulfate and chloride decreased obviously for the entire size range, with the
reduction ratio varying from 27% to 50% for sulfate and from 68% to 39% for chloride. In




contrast, the contribution of nitrate increased significantly for the entire size range, and the increase ratios at particle sizes > 317 nm (51-61%) were higher than those at particle sizes < 317 nm (31-55%). Additionally, the fraction of ammonium decreased at particle sizes < 317 nm, but increased slightly at particle sizes > 317 nm. The contribution of organics increased slightly at particle sizes < 317 nm or > 890 nm, while showed no significant change in the size range of 317-890 nm. These variations indicate the obvious changes in NR-PM$_{2.5}$ composition in winter Xi'an, which transfers from a sulfate-rich to a nitrate-rich composition with size-resolved characteristics.

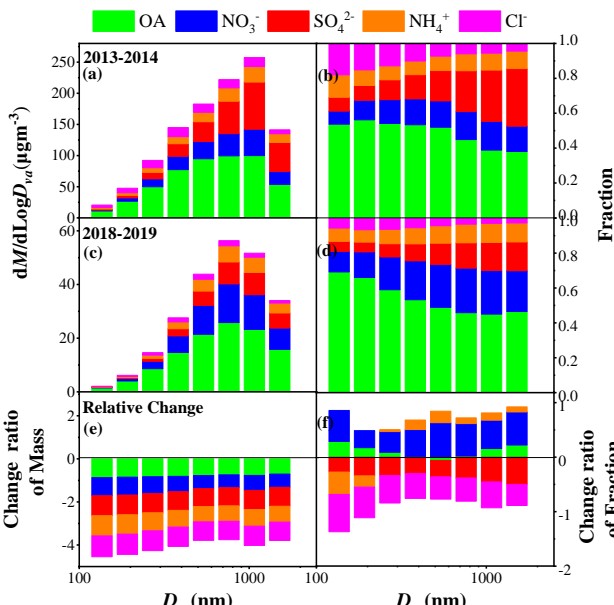

**Figure 1**. Average size distributions of mass concentrations and fractions of NR-PM$_{2.5}$ species (OA, NO$_3^-$, SO$_4^{2-}$, NH$_4^+$ and Cl$^-$) in winter 2013-2014 (a, b) and winter 2018-2019 (c, d) in Xi'an, as well as the relative change ratios of mass and fraction in 2018-2019 compared to 2013-2014 (e, f).

To elucidate the effects of changes in sulfate and nitrate, we further compared the size-resolved nitrate and sulfate contributions for different NR-PM$_{2.5}$ mass concentrations between 2018-2019 and 2013-2014. As shown in Fig. 2, during winter 2013-2014, nitrate and sulfate displayed similar contributions when the NR-PM$_{2.5}$ mass concentration was lower than 200 μg m$^{-3}$. As NR-PM$_{2.5}$ mass concentration increased from 200 μg m$^{-3}$ to 400-800 μg m$^{-3}$, the contribution of sulfate increased dramatically and maintained at a stable level, while the contribution of nitrate decreased linearly. In addition, under a constant NR-PM$_{2.5}$ mass concentration, the sulfate





contribution increased with the increase of particle size, while the decrease of nitrate contribution showed no obvious dependence on particle size. As a result, the ratio of $NO_3^-/SO_4^-$ decreased as NR-PM$_{2.5}$ mass concentration increased, with most values lower than 1, suggesting the more important role of sulfate than nitrate in haze pollution in winter 2013-2014. In comparison, during winter 2018-2019, the contributions of both nitrate and sulfate tended to increase as NR-PM$_{2.5}$ mass concentration increased across the entire mass concentration range. The increase of sulfate contribution was also more pronounced at larger particle sizes, while the increase of nitrate contribution had no obvious dependence on particle size, similar to that in 2013-2014. The ratio of $NO_3^-/SO_4^-$ was higher than 1 in the entire NR-PM$_{2.5}$ mass concentration range and entire particle size range, suggesting the obviously increased contribution of nitrate to haze pollution in 2018-2019. However, the ratio of $NO_3^-/SO_4^-$ showed a slightly decreased trend as NR-PM$_{2.5}$ increased, and decreased slightly as particle size increased at a constant NR-PM$_{2.5}$ mass concentration. This implies that although nitrate showed a higher contribution than sulfate in haze pollution in 2018-2019, the formation of sulfate was more efficient than nitrate under high NR-PM$_{2.5}$ mass concentrations, especially at larger particle sizes. This is similar to a recent study by Xie et al. (2023) where lower $NO_3^-/SO_4^-$ ratio was observed in serious pollution than in moderate pollution in Hohhot, China.

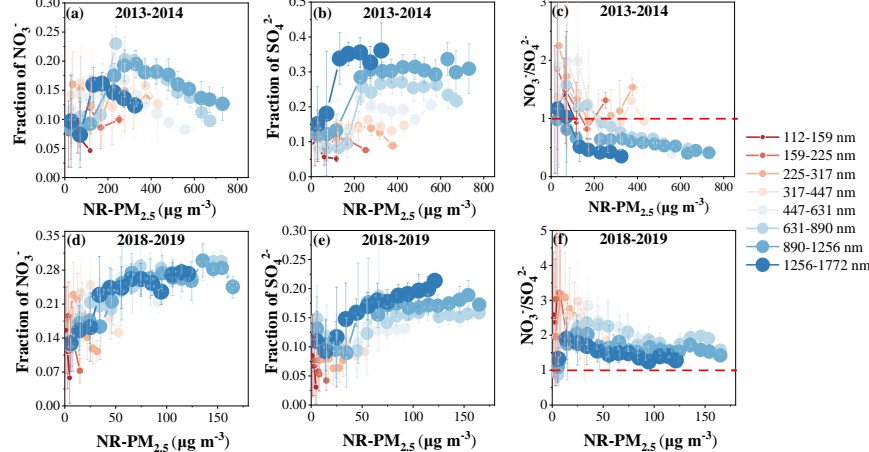

**Figure 2**. Variations of the fractions of nitrate and sulfate, as well as the ratio of $NO_3^-/SO_4^-$ at different NR-PM$_{2.5}$ levels in winter 2013-2014 (a, b, c) and winter 2018-2019 (d, e, f), respectively.

### 3.2 Size-resolved evolution of aerosol water uptake

The changes in aerosol composition especially inorganic composition also induce changes in aerosol water uptake. The size-resolved ALWC modelled by ISORROPIAII for 2018-2019 and 2013-2014 winters were compared (Fig. S3). Consistent with the size distribution of NR-PM$_{2.5}$,



ALWC also peaked at 631-890 nm in 2018-2019 and 890-1256 nm in 2013-2014. The ALWC in 2018-2019 was ~ 5-15 times lower than that in 2013-2014 across the entire size range, due to the ~ 4-11 times lower inorganic mass concentrations in 2018-2019. In addition, the frequency of RH > 60% in 2018-2019 (49%) was lower than that in 2013-2014 (64%) (Fig. S4). The less frequent occurrence of high RH together with the much lower inorganic mass concentration in 2018-2019 resulted in a lower ALWC than that in 2013-2014.

The mass growth factor of inorganic aerosol ($G_{mi}$), affected primarily by mass fractions of inorganic species rather than absolute mass, was further used to evaluate the response of aerosol water uptake to the changes in inorganic composition. As shown in Fig. 3, $G_{mi}$ showed an exponential increase with RH across the entire size range both in 2018-2019 and 2013-2014. The relatively size-dependent and RH-dependent changes of $G_{mi}$ in 2018-2019, compared to 2013-2014, was further analyzed (Fig. 3b). $G_{mi}$ in 2018-2019 increased relative to 2013-2014 in most cases, especially at larger particle sizes and higher RH conditions, with the highest increase ratio reaching 5-35% for particles > 317 nm at RH > 90%. However, decreased $G_{mi}$ in 2018-2019 relative to 2013-2014 was also occurred in some cases, in particular at small particle sizes and lower RH conditions. The variable changes in $G_{mi}$ were further suggested to be associated with the complicated changes in inorganic composition between 2018-2019 and 2013-2014 (Fig. S5 and S6).

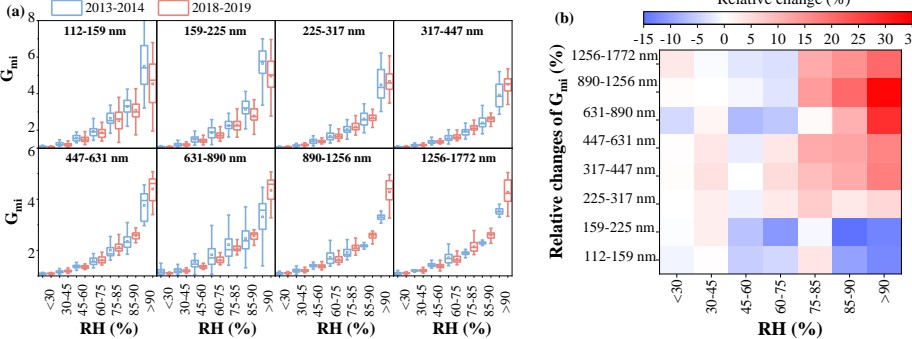

**Figure 3**. Variations of $G_{mi}$ as functions of RH under different size ranges between winter 2013-2014 and winter 2018-2019 in Xi'an (a), and relative changes of $G_{mi}$ in winter 2018-2019 compared to that in winter 2013-2014 across different particle sizes and different RH ranges (b).

At small particle sizes such as 112-225 nm, sulfate and nitrate in total contributed a lower mass fraction (~40-45%) to inorganics in 2013-2014, in which the sulfate and nitrate could be fully neutralized by ammonium, and a large excessed ammonium was further combined with chloride.



The important contribution of ammonium chloride to water uptake and its much higher hygroscopicity than other ones led to higher $G_{mi}$ in 2013-2014 at small particle sizes. This is similar to a recent observation in Delhi, India, where ammonium chloride was the largest contributor to aerosol water due to higher fraction of chloride, but lower fractions of sulfate and nitrate (Chen et al., 2022). Compared to 2013-2014, the increased sulfate and nitrate, but

decreased chloride fractions in 2018-2019 led to the increased contributions of ammonium sulfate and ammonium nitrate, but decreased contribution of ammonium chloride to water uptake, which then resulted in the decreased $G_{mi}$ in 2018-2019 at small particle sizes.

At larger particle sizes, however, sulfate and nitrate in total dominated inorganics (> 60%) in most cases both in 2013-2014 and 2018-2019, in which ammonium sulfate, ammonium

hydrogen sulfate and ammonium nitrate should be the main water-absorbing species. Under this condition, distinctly increased contribution of ammonium nitrate to water uptake in 2018-2019 will lead to increased $G_{mi}$, due to the higher hygroscopicity of ammonium nitrate than ammonium sulfate (Topping et al., 2005; Petters and Kreidenweis, 2007). Again, some cases with decreased $G_{mi}$ in 2018-2019 compared to 2013-2014 at larger particle sizes were associated

with a significant reduction in sulfate and nitrate contributions in 2013-2014, that is, the increased contribution of ammonium chloride to aerosol water (Fig.S5). Further analysis shows the cases with increased $G_{mi}$ in 2018-2019 compared to 2013-2014 were mainly related to enhanced contributions of both nitrate and ammonium (Fig. S6). These results highlight the importance of ammonium and chloride in impacting water uptake, in addition to commonly

recognized sulfate and nitrate.

### 3.3 Size-resolved changes of POA and SOA sources

We performed OA source apportionment in each size bin. Two OA sources including POA and SOA were resolved in winter 2013-2014, while three OA sources including FFOA, BBOA, and SOA were resolved in winter 2018-2019. The fewer OA factors resolved in this study than those

in Elser et al. (2016) and Duan et al. (2022) were due to the lower resolution of size-resolved mass spectrometric data (unit mass resolution, UMR) used in this study compared to the bulk mass spectrometric data (high resolution, HR) used in Elser et al. (2016) and Duan et al. (2022). Here, we defined FFOA+BBOA as POA in winter 2018-2019 for better comparison (Fig. 4). As particle size increased from 112 nm to 1772 nm, the fraction of SOA in total OA increased

from 27% to 40% in 2018-2019 and from 15% to 43% in 2013-2014. Both the mass concentrations of POA and SOA in 2018-2019 decreased relative to 2013-2014 across the entire size range. Specifically, the reduction ratio of POA in the smaller size mode was slightly higher than that in the larger size mode, which decreased from 85% to 65% as particle size increased from 112 nm to 1772 nm. In comparison, the reduction ratio of SOA showed no obvious

dependence on the particle size, fluctuating around 70% as particle size increased from 112 nm to 1772 nm. As a result, compared to 2013-2014, the fraction of SOA in total OA in 2018-2019





mainly increased at smaller sizes of 112-317 nm, while showed no obvious changes at larger sizes of 317-1772 nm due to the similar decrease ratio between POA and SOA concentrations.

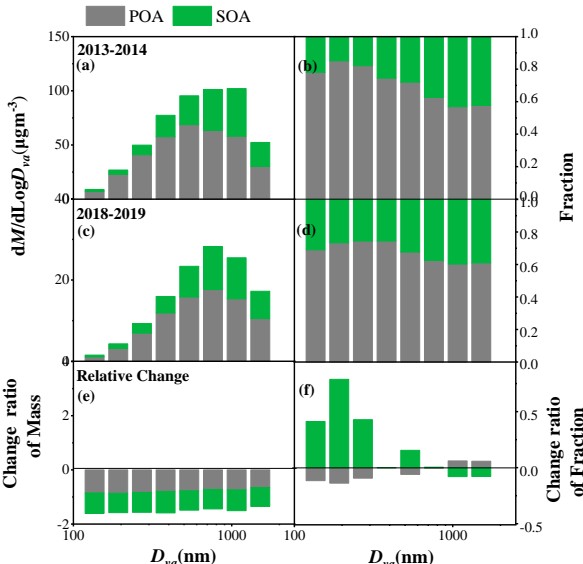

**Figure 4**. Average size distributions of mass concentrations and fractions of POA and SOA factors in winter 2013-2014 (a, b) and winter 2018-2019 (c, d) in Xi'an, and the relative change ratios of mass and fraction in 2018-2019, compared to 2013-2014 (e, f).

### 3.4 Enhanced influence of aerosol water on SOA formation

Meteorological parameters such as RH, temperature, SR, as well as Ox and aerosol water play important roles in the photochemical and aqueous-phase processes of SOA (Huang et al., 2019; Kuang et al., 2020). We used both variable importance and SHAP methods in random forest model to evaluate the size-resolved impacts of these factors in SOA formation between winter 2018-2019 and winter 2013-2014. Normalized aerosol water (ALWC/NR-PM$_{2.5}$, Liu et al., 2021) and SOA (the mass fraction of SOA in total OA) were applied to eliminate the effect of simultaneous increases or decreases in the absolute concentrations of aerosol species. Note the contribution of organics to ALWC was not considered in ISORROPIA II modelling, which will lead to an underestimation of ALWC/NR-PM$_{2.5}$. However, as the fraction of organics in total NR-PM$_{2.5}$ changed less in winter 2018-2019 compared to winter 2013-2014, particularly at particle sizes of 225-1256 nm, the effects of organic changes on ALWC/NR-PM$_{2.5}$ should also be small (Fig. 4). The performance of the model results was assessed by the correlation between the predicted and observed values of the test data (Table S2), and only those size ranges with correlation efficiency (R$^2$) above 0.5 in both 2013-2014 and 2018-2019 were selected for comparison. Fig. 5 shows the relative importance of these factors in influencing SOA formation.





In winter 2013-2014, the SOA formation was mainly affected by $O_x$ and temperature, but showed a lower dependence on aerosol water with relative importance ranging from 19% to 22% across the size range of 225-1256 nm. However, enhanced impact of aerosol water on SOA formation was observed during winter 2018-2019. The relative importance of aerosol water increased from 21% to 51% as particle size increased from 225 nm to 1256 nm, and became the greatest influence factor on SOA formation for particle sizes of 447-1256 nm in winter 2018-2019.

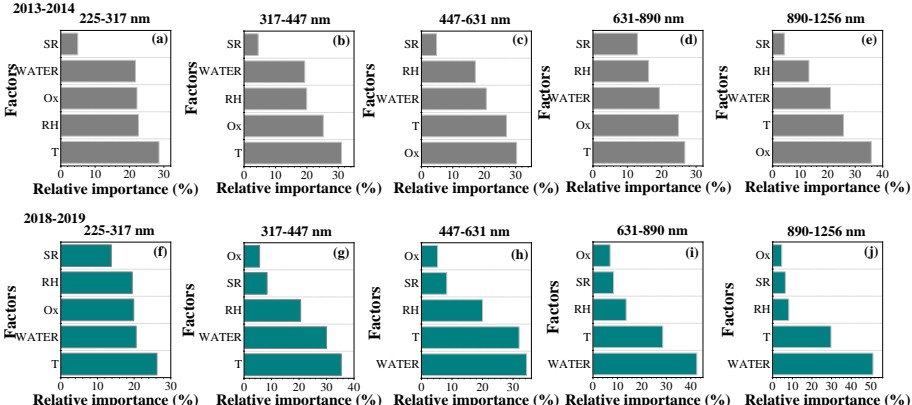

**Figure 5.** Random forest analysis for relative importance of factors impacting SOA formation during winter 2013-2014 (a-e) and winter 2018-2019 (f-j) in Xi'an.

Similarly, SHAP modelling results also show increased contribution of aerosol water to SOA formation in winter 2018-2019, compared to winter 2013-2014 (Fig. 6). In winter 2013-2014, a positive SHAP value was observed more frequently for $O_x$, and the SHAP value of temperature increased positively as temperature increased across the size range of 225-1256 nm. However, aerosol water displayed an SHAP value close to zero or slightly negative in most cases. This suggests that SOA was mainly formed from photochemical oxidation in winter 2013-2014 (Elser et al., 2016). Compared to winter 2013-2014, $O_x$ showed less contribution to SOA formation with a close to zero SHAP value more frequently in winter 2018-2019. In contrast, aerosol water contributed positively to SOA formation in most cases and the SHAP value increased as aerosol water increased, especially at larger particle sizes, suggesting enhanced positive impact of aerosol water on SOA formation in winter 2018-2019. We suppose that the majority of enhanced aerosol water uptake in winter 2018-2019 at larger particle sizes and high RH might facilitate the efficient gas-particle partitioning of water-soluble organic compounds, and thus efficient aqueous-phase formation of SOA (Lv et al., 2023). Additionally, a most recent study by Liu et al. (2023) revealed that urban aerosol particles could exist in a liquid state at lower RH levels with increased nitrate fraction, and the diffusion coefficient was



significantly enhanced in nitrate-dominated particles, making them key seeds for secondary aerosol formation through multiphase reactions. This might be another reason for the enhanced aqueous-phase formation of SOA in winter 2018-2019, compared to winter 2013-2014 in our study. These results highlight the importance of aerosol water that links the hygroscopicity of inorganic species to multiphase processes of secondary organics, and imply the potential

influence of inorganic ions on the SOA formation in urban aerosol.

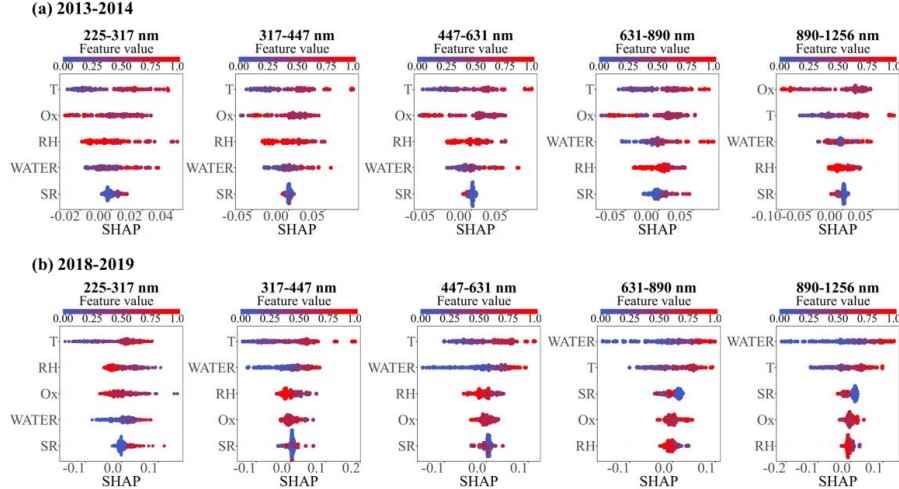

**Figure 6.** SHAP values for the analysis of the importance of each factor in SOA formation between winter 2013-2014 (a) and winter 2018-2019 (b) for sizes of 225-1256 nm.

**4. Conclusions**

Size-resolved NR-PM$_{2.5}$ composition and OA sources were compared between winters 2018-2019 and 2013-2014 in Xi'an. The NR-PM$_{2.5}$ composition changed significantly from a sulfate-rich to a nitrate-rich characteristic from winter 2013-2014 to winter 2018-2019, especially at larger particle sizes. The size-resolved changes in chemical composition resulted in size-

dependent changes of aerosol water uptake. Compared to 2013-2014, $G_{mi}$ in 2018-2019 increased in most cases, especially at larger particle sizes and higher RH conditions, with the highest increase ratio reaching 5-35% for particles > 317 nm. In particular, the important role of ammonium and chloride in impacting water uptake was emphasized, in addition to sulfate and nitrate. The random forest analysis showed enhanced relative importance of aerosol water

in impacting SOA formation from 2013-2014 to 2018-2019, especially at larger particle sizes. Ox and temperature showed dominant and positive contribution to SOA formation in 2013-2014, while aerosol water became the leading factor contributing positively to SOA formation in 2018-2019, implying an enhanced aqueous-phase formation of SOA. This study highlights the key role of aerosol water uptake, which links inorganic species and organic compounds in



their secondary processes. The potential impact of inorganic ions, especially nitrate, in the multiphase processes of SOA and the key mechanism need to be further clarified in urban nitrate-dominant particles.

*Data availability*. The data used in this study are archived at the East Asian Paleoenvironmental Science Database, National Earth System Science Data Center, National Science & Technology
Infrastructure of China: https://doi.org/10.12262/IEECAS.EAPSD2024001 (Duan et al., 2024).

*Supplement*. The Supplement related to this article is available online at

*Author contributions*. RJH designed the study. JD, YW, HBZ and YFG conducted the field observation. JD, RJH analyzed the data, with help from YW, WX and CSL. JD and RJH wrote the manuscript, and WH, JO, DC, and CO all commented on and discussed the manuscript.

*Competing interests*. The authors declare that they have no conflict of interest.

**Acknowledgements.** This work was supported by the National Natural Science Foundation of China (NSFC) under Grant No. 41925015, the Strategic Priority Research Program of Chinese Academy of Sciences (XDB40000000), the Key Research Program of Frontier Sciences from the Chinese Academy of Sciences (ZDBS-LY-DQC001), the Natural Science Basic Research
Program of Shaanxi Province (2023-JC-QN-0319), and the National Key Research & Development Program (2023YFC3705503).

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
