# Peer review of "Measurement Report: Size-resolved secondary organic aerosol formation modulated by aerosol water uptake in wintertime haze"

_EGUsphere, 2024_

## Author Comment (AC1)

The authors thank the referees to review our manuscript and particularly for the valuable comments and suggestions that are very helpful in improving the manuscript. We provide below point-by-point responses to the referees' comments. We also have made most of the changes suggested by the referees in the revised manuscript.

**Response to Referee #1**

By analyzing the size-resolved chemical composition of non-refractory fine particulate matter obtained at winter 2013-2014 and winter 2018-2019, this study investigated the potential effects of inorganics changes on aerosol water uptake and secondary organic aerosol formation. They found the increased water uptake at larger particles as the aerosol chemical profile shifted from a sulfate-rich to a nitrate-rich. Further, they reported that the enhanced aerosol water uptake at larger particle sizes resulted by the changed aerosol chemical profile, would facilitate the efficient aqueous-phase SOA formation, which provided an interesting perspective to evaluate the role of inorganics and organics in their multiphase processes. Yet, there are some issues that need to be addressed for further improving this work.

Line 24-25, it is better to give a specific value here for quantify the "higher relative-humidify".
**Response:** We thank the referee's suggestion. In the revised manuscript lines 24-25, the sentence has been updated, now reading: "……with the highest increase ratio reaching 5-35% at larger particle sizes and higher relative-humidity (RH > 70%)".

Line 28-30, it is unclear what the implicit relation between SHAP value and the aerosol water.
**Response:** We thank the referee's comment. In our study, the SHAP method was employed to evaluate the impact of each factor (including RH, T, Ox, solar radiation, and aerosol water) on SOA formation. A higher SHAP value of aerosol water indicates its increased importance and contribution to SOA formation. To be more accurate and clear, we have updated the sentence in the revised manuscript lines 28-33, which now reads: "……Aerosol water exhibited a significant contribution to SOA formation during winter 2018-2019, particularly at larger particle sizes. The SHAP value of aerosol water increased alongside higher levels of aerosol water, indicating an enhanced contribution of aerosol water to SOA formation. This implies……".

Line 89-91, two kinds of instruments were used for the measurement of NR-PM2.5 and its size-resolved chemical composition. Is there any difference for quantification of organic and inorganic components? A brief explanation should be provided here.
**Response:** We thank the referee's comment and suggestion. In our study, a high resolution-time of flight-aerosol mass spectrometer (HR-ToF-AMS, Aerodyne Research Inc.) in 2013-2014 and a soot particle-long time of flight-aerosol mass spectrometer (SP-LToF-AMS, Aerodyne Research Inc.) in 2018-2019 were employed, respectively. The SP-LToF-AMS operated as a

standard LToF-AMS in "laser off" mode in our study, exclusively measuring non-refractory PM$_{2.5}$ (NR-PM$_{2.5}$) species, aligning with the HR-ToF-AMS measurements. The LToF-AMS is an updated version of HR-ToF-AMS, employing identical quantification and measurement principles for mass concentration and aerodynamic size distribution of NR-PM$_{2.5}$ species, and utilizing the same software SQUIRREL (for UMR analysis) and PIKA (for HR analysis) for data analysis. Notably, the LToF-AMS features a longer ToF mass spectrometer chamber to improve mass resolution and facilitate better separation of closely positioned fragments in high-resolution mass spectrum analysis. Additionally, it incorporates a multiplexed chopper and efficient PToF (ePToF) mode to achieve a higher sizing duty cycle during particle sampling (50% compared to 2% in the standard chopper and PToF mode). To ensure data comparability, ionization efficiency (IE) calibration and size calibration utilizing NH$_4$NO$_3$ were conducted for both HR-ToF-AMS and LToF-AMS during the 2013-2014 and 2018-2019 campaigns, respectively. Similarly, various versions of aerosol mass spectrometers such as HR-AMS, LToF-AMS, and ToF-ACSM (a simplified AMS model with reduced complexity and performance) have been collectively utilized in many previous studies for comprehensive data analysis and comparison across diverse locations and seasons (Xu et al., 2015; Chen et al., 2022; Zheng et al., 2021, 2023).

In the revised manuscript lines 92-107, a brief explanation for the two instruments has been added as follows:

"……The size-resolved NR-PM$_{2.5}$ composition was analyzed using a high resolution-time of flight-aerosol mass spectrometer (HR-ToF-AMS, Aerodyne Research Inc.) in PToF mode during the winter of 2013-2014, and a soot particle-long time of flight-aerosol mass spectrometer (SP-LToF-AMS, Aerodyne Research Inc.) in efficient PToF (ePToF) mode in the winter of 2018-2019. In our study, the SP-LToF-AMS operated as a standard LToF-AMS in "laser off" mode, exclusively measuring non-refractory aerosols, aligning with the HR-ToF-AMS measurements. The LToF-AMS is an upgraded version of the HR-ToF-AMS, employing the same quantification principle for non-refractory organic and inorganic components while featuring an extended ToF mass spectrometer chamber to enhance mass resolution and a multiplexed chopper in ePToF mode for a higher sizing duty cycle during particle sampling. To ensure data comparability, ionization efficiency (IE) calibration and size calibration utilizing NH$_4$NO$_3$ were conducted for both instruments during the 2013-2014 and 2018-2019 campaigns, respectively. Detailed information on instrument operation and calibration can also be found in previous studies (Elser et al., 2016; Duan et al., 2022)".

Line 97-99, the experimentally determined RIEs and standard RIEs were used for different components, why?

**Response:** We thank the referee's comment. During the AMS calibrations, ionization efficiency (IE) calibration using NH$_4$NO$_3$ was first conducted to convert ion signals to mass concentrations. RIE is the relative ionization efficiency of species, compared to nitrate (Jimenez et al., 2003).

We conducted $NH_4NO_3$ and $(NH_4)_2SO_4$ calibrations during both the 2013-2014 and 2018-2019 campaigns, and obtained RIEs for ammonium and sulfate, thus experimentally determined RIEs of ammonium and sulfate were used in our campaigns. However, since there were no RIE calibrations for other species during our campaigns, default RIE values of 1.4, 1.1, and 1.3 were employed for other species such as organics, nitrate, and chloride, respectively. These default values are typically used in AMS ambient concentration calculations (Canagaratna et al., 2007; Zhang et al., 2014; Xu et al., 2020). It is worth noting that the RIE of nitrate is greater than 1 because the calibrations monitor $NO^+$ and $NO_2^+$, which together account for only 90% of the total ion signal from the $NO_3$ group (Canagaratna et al., 2007).

To be more accurate, in the revised manuscript lines 112-119, the sentence has been updated as follows: "……and from SP-LToF-AMS in 2018-2019. Based on the $NH_4NO_3$ and $(NH_4)_2SO_4$ calibrations, experimentally determined RIEs of 1.48 and 3.37 in 2013-2014, and 1.30 and 3.70 in 2018-2019, were used for sulfate and ammonium, respectively. In comparison, default RIEs of 1.4, 1.1, and 1.3 were used for organics, nitrate, and chloride, respectively, according to previous studies (Canagaratna et al., 2007; Zhang et al., 2014; Xu et al., 2020)".

Line 174-175, in winter 2013-2014, chloride are obviously concentrated on the smaller particle size, while it maintained a relatively smaller and stable contribution in winter 2018-2019. Please explain it.

**Response:** We thank the referee's comment. Biomass burning and coal combustion are two significant primary emission sources of chloride in winter, typically found in the smaller particle size mode (Xu et al., 2016, 2020; Ye et al., 2017). Although CCOA and BBOA were not completely distinguished in our UMR data analysis, according to the source apportionment results for bulk OA during the winter of 2013-2014 (Elser et al., 2016), CCOA and BBOA together contributed 56% to total OA on reference days and 49% to total OA on haze days, potentially accounting for the elevated chloride levels in smaller particle sizes. In contrast, the combined contribution of CCOA and BBOA within bulk OA decreased to 21% in the winter of 2018-2019 (Duan et al., 2022), indicating a decline in primary chloride sources. Consequently, chloride exhibited a relatively minor and consistent contribution across the size ranges in winter 2018-2019 compared to that in 2013-2014.

In the revised manuscript lines 210-213, we have added an explanation, and the sentence now reads: "…… and from 68% to 39% for chloride. The significant reduction in chloride contribution, particularly in smaller particle sizes, indicates a decrease in primary chloride emission sources during winter from 2013-2014 to 2018-2019. In contrast……".

Line 215. As showed in Figure 2, the mass range of NR-PM2.5 in 2018-2019 are much smaller than that in 2013-2014, it is better to explore and compare the variations in the fractions of nitrate and sulfate in the same mass range of NR-PM2.5.

**Response:** We thank the referee's suggestion. Accordingly, we explored and compared the

variations in the fractions of nitrate and sulfate in the same mass range of NR-PM$_{2.5}$ between 2013-2014 and 2018-2019. However, as shown in Fig. R1, the data from 2013-2014 display no prominent trends and exhibit fluctuations in a disorderly manner. Given that the mass range of NR-PM$_{2.5}$ in 2018-2019 is considerably smaller than that in 2013-2014, the data from 2013-2014 within the equivalent mass range of NR-PM$_{2.5}$ in 2018-2019 contribute a significantly lower proportion to the total data for each size range in 2013-2014. When these data were further categorized into different NR-PM$_{2.5}$ mass bins for analysis, only minimal data points (sometimes fewer than 5 data points) were present in each mass bin. This led to larger error bars on the points, diminishing the credibility and causing fluctuations in the trends. Furthermore, due to the small percentage of these data within the total dataset for 2013-2014, comparing the trends in this range would overlook the information regarding variations in the nitrate and sulfate fractions and their relative roles during haze pollution episodes with higher NR-PM$_{2.5}$ mass concentrations in more dominant time periods in 2013-2014. Therefore, we examined and compared the variations in the fractions of nitrate and sulfate across the entire NR-PM$_{2.5}$ mass range for 2013-2014 and 2018-2019, respectively.

[Figure]

Fig. R1 The variations in the fractions of nitrate, sulfate and the ratio of nitrate/sulfate in the same mass range of NR-PM$_{2.5}$ between 2013-2014 and 2018-2019.

Line 300-305, I agree that the fraction of organics in total NR-PM2.5 changed less in winter 2018-2019 compared to winter 2013-2014. However, the fraction of water-soluble organics, which contributed more to ALWC related to total organics, maybe varied a lot.

**Response:** We thank the referee for pointing this out. We agree with the referee that the fraction of water-soluble organics contributing more to ALWC maybe vary a lot between 2013-2014 and 2018-2019. However, as our study did not measure the chemical composition of organics, information regarding changes in organic composition, especially the variations in the water-soluble organics fraction, was not available. In response to referee 2's comment below,

ALWC$_{org}$ (ALWC attributed to organics estimated assuming $\kappa_{org}$ = 0.06) demonstrated lower contributions to total ALWC (< 20%) across the entire size range in both 2013-2014 and 2018-2019, indicating that inorganic aerosols were the primary hygroscopic species. Therefore, our focus primarily centered on ALWC influenced by changes in inorganic species in this study. Nevertheless, we thank the referee's emphasis on this aspect. Given that the chemical composition of organics, particularly water-soluble organics, may have undergone significant alterations in recent years, impacting organic hygroscopicity, ALWC contribution, and aqueous-phase chemistry, these factors should be investigated in future studies through comprehensive measurements and detailed characterization of organic composition changes.

To be more accurate, the discussion mentioning "However, as the fraction of organics in total NR-PM$_{2.5}$ changed less in winter 2018-2019 compared to winter 2013-2014, particularly at particle sizes of 225-1256 nm, the effects of organic changes on ALWC/NR-PM$_{2.5}$ should also be small (Fig. 4)" has been removed from the revised manuscript.

Line 319-325, a clear explanation for the intend implication of theSHAP value should provide first before the discussion on SOA formation.

**Response:** We thank the referee's suggestion, and a clear explanation of the methodology for SHAP analysis and the intend implication of the SHAP value have been added in section 2.5 before the results and discussion in the revised manuscript lines 176-190 as follows:

"......Moreover, the SHapley Additive explanation (SHAP) algorithm was applied to evaluate the importance of each variable in predicting SOA. SHAP is a novel interpretable method for machine learning models based on the concept of Sharpley value in game theory, which can help interpret the complex model predictions (nonlinear fitting) and quantify the input variables' contributions to a single prediction (SOA) with physical significance (Lundberg et al., 2020). The functional expression of SHAP can be defined as follows:

$$f(x_i)=\phi_0(f,x)+\sum_{i=1}^{M}\phi_j(f,x_i) \qquad (3)$$

where $f(x_i)$ denotes the predicted value produced for each sample $(x_i)$ with M features. $\phi_0(f,x)$ serves as the base value representing the anticipated output value of the RF model across the dataset. Additionally, $\phi_j(f,xi)$ represents the SHAP value delineating the influence of feature j in the sample $(x_i)$ on the prediction of that specific sample (Lundberg et al., 2020)".

Meanwhile, in the revised manuscript lines 357-361, the intend implication of the SHAP value in fig. 6 has also been explained firstly before the discussion on SOA formation, which now reads: "Similarly, ......The variables are arranged in descending order of importance from top to bottom. A negative or positive SHAP value signifies that the variable contributes more, while a SHAP value closer to zero indicates less contribution to the SOA formation.......".

Line 329-332, as the particle mass concentration had been largely reduced in 2018-2019, the

gas-particle partitioning of water-soluble organic compounds would be also suppressed as the aerosol surface been decreased. What about the aerosol acidity with increased nitrate fraction? And what the role of aerosol acidity on the SOA formation through multiphase reactions?

**Response:** We thank the referee's comment. We agree with the referee that the total aerosol surface decreased as the particle mass concentration largely reduced in 2018-2019. Nevertheless, our primary focus in this study was on examining the impact of the increased nitrate fraction and aerosol water uptake on the gas-particle partitioning of water-soluble organic compounds and subsequent chemical processes involved in SOA formation. According to Lv et al. (2023), a shift in haze pollution from being sulfate-dominant to nitrate-dominant could lead to increased aerosol hygroscopicity and enhance the gas-to-particle-phase partitioning of water-soluble organics. Therefore we suppose that the majority of enhanced aerosol water uptake in winter 2018-2019 at larger particle sizes and high RH in our study might also facilitate the efficient gas-particle partitioning of water-soluble organic compounds, and thus efficient aqueous-phase formation of SOA.

We thank the referee for highlighting the significant aspect of aerosol acidity and its potential impact on SOA formation. In this study, as measurements of gas species such as ammonia ($NH_3$), nitric acid ($HNO_3$), and hydrochloric acid (HCl) were not available, ISORROPIA II was operated in reverse mode (utilizing only aerosol-phase composition as input) to determine the aerosol water concentrations. However, when attempting pH calculation using the ISORROPIA II outputs in reverse mode, serious inaccuracies arose with pH values peaking at much lower (-2-2) or higher (7-10) ranges due to strong influences from ionic measurement errors. Consequently, reverse-mode calculations should be avoided in pH determination (Hennigan et al., 2015; Song et al., 2018). Moreover, gaseous $NH_3$, which was lacked in this study, plays a critical role in pH computation using the forward mode (requiring total gas and particle phase inputs) recommended in previous studies (Wang et al., 2016; Song et al., 2018). Therefore, given the various limitations associated with pH computation, the change in aerosol acidity and its impacts on SOA formation were not discussed in our study.

Nevertheless, previous studies have indicated that the particle pH rises with an increase in the nitrate fraction in various urban cities such as Beijing and Shanghai in recent years (Xie et al., 2020; Lv et al., 2023). Lv et al. (2023) specified that during sulfate-dominant haze periods, the uptake behavior of water-soluble organic compounds in the gas phase (WSOCg) was predominantly determined by aerosol pH, while during nitrate-dominant haze periods, it was influenced by ALWC. The higher pH and ALWC observed during nitrate-dominant events led to a significant rise in WSOCg partitioning and enhanced SOA formation. Additionally, the acidity of aqueous atmospheric solutions plays crucial roles in key chemical mechanisms and kinetics in multiphase reactions (Tilgner et al., 2021). These results suggest that the notable alterations in aerosol water content and aerosol acidity in recent years, driven by evident changes in aerosol composition, could substantially impact multiphase reactions and SOA formation. This aspect should be further emphasized and explored in future work.

**Response to Referee #2**

This study investigated the factors influencing SOA formation at different particle sizes. Data from two observations (2013–2014, 2018–2019) in Xi'an in winter revealed that the composition of inorganic aerosols changed significantly from sulfate-rich to nitrate-rich at different particle sizes. This transition resulted in changes in aerosol water uptake. Further analysis using random forest and the Shapley additive explanation algorithm (SHAP) elucidated the relative significance of aerosol water in SOA formation, particularly at larger particle sizes. This finding implies that the enhancement of aerosol water uptake at larger particle sizes and high RH may contribute to liquid-phase SOA formation. Before this article was published, there may have been some issues that needed to be corrected as follows:

General comments

The detailed methodology of PMF analysis as a function of particle size is not clear. Is 3D-matrix or 2D matrix PMF used? How about the time series of the size-resolved PMF factor? Detailed information on how to obtain the best solution for the size-resolved PMF shall be shown.

**Response:** We thank the referee's comment and suggestion, and the detailed methodology and information of PMF analysis have been added in the revised manuscript accordingly. In our study, the OA mass spectra within the range of 113-1772 nm were binned into 8 size ranges, and 2D matrix PMF was utilized to analyze the OA mass spectra in each size bin for both 2013-2014 and 2018-2019. The time series of the size-resolved PMF factors for 2013-2014 and 2018-2019 are shown in Fig. R2 and Fig. R3, respectively, which have also been added to the revised supplement.

In the revised manuscript lines 141-153, the paragraph about PMF analysis has been updated as follows:

"2D matrix PMF was utilized to analyze the OA mass spectra in each size bin for both 2013-2014 and 2018-2019 campaigns. Briefly……The methodology of size-resolved PMF analysis and selection of best solutions were detailed in the supplement (SI-text). Two OA sources including primary organic aerosol (POA) and SOA were resolved in each size bin in 2013-2014, and three OA sources including fossil-fuel-related OA (FFOA), biomass burning OA (BBOA) and SOA were resolved in each size bin in 2018-2019 (Fig. S1-S4)……"

In the revised supplement, the detailed information on how to obtain the best solution for the size-resolved PMF has been added as follows:

"SI-text: OA source apportionment

For OA mass spectra in each size bin during winter campaigns of 2013-2014 and 2018-2019, unconstrained PMF runs with varying factor numbers were conducted. The optimal solution for the size-resolved PMF was determined based on the principle that reducing the number of factors results in the mixing of different sources, while increasing the number of factors leads

to factor splitting or mixing again and the emergence of non-meaningful factors.

Briefly, for the PMF results in each size bin in 2013-2014, the 2-factor solution resolved a POA source with prominent signal peaks at fragments such as m/z 55 (mainly $C_4H_7^+$), m/z 57 (mainly $C_4H_9^+$), and m/z 60 (mainly $C_2H_4O_2^+$), while a much lower signal at m/z 44 (mainly $CO_2^+$, a typical fragment in aging or secondary OA sources). These features align with typical characteristics of POA profiles reported in previous studies at various urban sites (Ng et al., 2010, 2011; Hu et al., 2016). In contrast, the second factor exhibited significantly lower signals for primary fragments like m/z 55, 57, and 60, but displayed a dominant signal peak at m/z 44, indicating its aging or secondary nature (Ng et al., 2011). Consequently, this factor was classified as a SOA source (Fig. S1). Upon increasing the factor number to 3, no distinct POA categories like HOA, COA, or BBOA were further resolved. However, the SOA factor was split into two factors, with signals at m/z 29 and m/z 44 divided in Factor 2, and signals at m/z 28 and m/z 43 divided in Factor 3, respectively (Fig. S5). Consequently, the 2-factor solution was selected as the best performance, and two OA sources, comprising POA and SOA, were identified in each size bin in 2013-2014.

For the PMF results in each size bin in 2018-2019, Factor 2 in the 2-factor solution was evidently a mixture of m/z 44 and numerous primary ions of m/z > 50, notably m/z 60 originating from BBOA (Fig. S6). This observation suggests that the 2-factor solution did not clearly distinguish between SOA and POA. Upon increasing the factor number to 3 (Fig. S3), two primary sources with much lower contribution from m/z 44, and one secondary source with dominant signal peaks at m/z 44 were resolved. Specifically, as a tracer ion for biomass burning sources, the signal of m/z 60 appeared primarily in Factor 2, with no significant mixing observed in Factor 1 and Factor 3. This indicates that Factor 2 represents a clean BBOA source identified through PMF analysis (Cubison et al., 2011). Factor 1, another primary source unaffected by m/z 60 interference, was characterized by prominent hydrocarbon ion series of $C_nH_{2n-1}$ and $C_nH_{2n+1}$, defining it as a fossil-fuel-related OA (FFOA) according to previous studies (Sun et al., 2016, 2018). In comparison, Factor 3 exhibited a dominant signal peak at m/z 44, which was defined as a SOA source. The 4-factor solution was further analyzed to determine if increasing the number of factors could resolve more refined primary or secondary sources. As shown in Fig. S7, increasing the factor number to 4 resulted in newly generated factors frequently showing significant mixing, such as a mixture of BBOA and FFOA in the smaller size ranges and a blend of secondary and primary sources, especially m/z 60, in the larger size ranges. Moreover, continuing to increase the factor number to 5 did not alleviate the mixing issues, instead, other non-meaningful factors were observed (Fig. S8). Therefore, the 3-factor solution was selected as the best performance, and three OA sources, comprising FFOA, BBOA and SOA, were identified in each size bin in 2018-2019".

In addition, Fig. R2-R7 shown below have been added to the revised supplement as Fig. S2, Fig. S4, and Fig. S5-S8 accordingly.

[Figure]

Fig. R2 Time series of (a1-h1) POA, (a2-h2) SOA under different size ranges of 112-159 nm, 159-225 nm, 225-317 nm, 317-447 nm, 447-631 nm, 631-890 nm, 890-1256 nm, and 1256-1772 nm, respectively, in winter 2013-2014 in Xi'an.

[Figure]

Fig. R3 Time series of (a1-h1) FFOA, (a2-h2) BBOA, and (a3-h3) SOA under different size ranges of 112-159 nm, 159-225 nm, 225-317 nm, 317-447 nm, 447-631 nm, 631-890 nm, 890-1256 nm, and 1256-1772 nm, respectively, in winter 2018-2019 in Xi'an.

[Figure]

Fig. R4 Mass spectra of three-factor PMF results under different size ranges of 112-159 nm, 159-225 nm, 225-317 nm, 317-447 nm, 447-631 nm, 631-890 nm, 890-1256 nm, and 1256-1772 nm, respectively, in winter 2013-2014 in Xi'an.

[Figure]

Fig. R5 Mass spectra of two-factor PMF results under different size ranges of 112-159 nm, 159-225 nm, 225-317 nm, 317-447 nm, 447-631 nm, 631-890 nm, 890-1256 nm, and 1256-1772 nm, respectively, in winter 2018-2019 in Xi'an.

[Figure]

Fig. R6 Mass spectra of four-factor PMF results under different size ranges of 112-159 nm, 159-225 nm, 225-317 nm, 317-447 nm, 447-631 nm, 631-890 nm, 890-1256 nm, and 1256-1772 nm, respectively, in winter 2018-2019 in Xi'an.

[Figure]

Fig. R7 Mass spectra of five-factor PMF results under different size ranges of 112-159 nm, 159-225 nm, 225-317 nm, 317-447 nm, 447-631 nm, 631-890 nm, 890-1256 nm, and 1256-1772 nm, respectively, in winter 2018-2019 in Xi'an.

Are the fractions of SOA and POA in total OA based on sizes-resolved PMF consistent with the fractions obtained with MS data only? If not, what might cause the differences?

**Response:** We thank the referee for pointing this out, and we compared the size-resolved fractions of SOA and POA in total OA in our study with those obtained through bulk-MS PMF in Elser et al. 2016 (2013-2014) and Duan et al. 2022 (2018-2019), respectively. In 2013-2014, the fraction of POA in total OA ranged from 85% to 57%, while the fraction of SOA varied from 15% to 43% as the particle size increased from 113 nm to 1772 nm. This range of fractions align with the results in Elser et al. 2016 (with an average of 75% POA and 25% SOA). In comparison, in 2018-2019, the fraction of POA in total OA ranged from 73% to 60%, while the fraction of SOA varied from 27% to 40% as the particle size increased from 113 nm to 1772 nm. The size-resolved contribution of SOA is lower than that observed in Duan et al. 2022 (with an average of 44% POA and 56% SOA). This difference may be attributed to the lower mass resolution of the size-resolved UMR data used for PMF analysis in this study. HR-MS data were utilized and a secondary OOA-BB source was resolved in Duan et al. 2022, which on average contributed 23% to the total OA. However, for the size-resolved PMF results in this study, the OOA-BB source was not discerned and might have been mixed into other sources, leading to lower size-resolved SOA fractions compared to that resolved based on MS data in Duan et al. 2022.

Section 3.2 the organic aerosols account for a large fraction of total PM masses and the fraction increases with the particle sizes get higher. Thus, the aerosol water uptake contributed by organics cannot be ignored and shall be considered here. E.g., method in Guo et al. (2015).

**Response:** We thank the referee's suggestion. We agree with the referee that the organic aerosol account for a large fraction of total PM masses. According to the referee's suggestion, we calculated the ALWC contributed by organics ($ALWC_{org}$) following the method in Guo et al (2015):

$$W_{org} = \frac{OM}{\rho_{org}} \cdot \rho_w \cdot \frac{\kappa_{org}}{(100\%/RH - 1)}$$

where OM is the mass concentration of organics, $\rho_w$ is the density of water, $\rho_{org}$ is the density of organics ($\rho_{org} = 1.4 \times 10^3$ kg m$^{-3}$, Cerully et al., 2015), and $\kappa_{org}$ is the hygroscopicity parameter of organic aerosol.

Since the cloud condensation nuclei (CCN) measurements and calculation of $\kappa_{org}$, performed in Guo et al. (2015), were not available in our measurements, we adopted a $\kappa_{org}$ value of 0.06 based on previous CCN measurements (Gunthe et al., 2011; Wu et al., 2016) to estimate the $ALWC_{org}$. As shown in Fig. R8, $ALWC_{org}$ exhibited lower contributions to total ALWC ($<20\%$) across the entire size range during both 2013-2014 and 2018-2019, suggesting that inorganic aerosols were the predominant hygroscopic species. This result aligns with our previous study (Huang et al., 2020).

Given that inorganic aerosols contributed a dominant fraction than organics to aerosol water, and the utilization of $\kappa_{org} = 0.06$ in $ALWC_{org}$ calculation failed to capture the hygroscopicity variations between 2013-2014 and 2018-2019 due to chemical changes in organics, our primary focus in this study was on the ALWC mediated by changes in inorganic species.

Nonetheless, we appreciate the referee for highlighting this aspect. Considering that the chemical composition of organics, particularly water-soluble organics, may have undergone significant changes in recent years, impacting the hygroscopicity of organics, contributing to ALWC, and influencing aqueous-phase chemistry, these aspects should be explored in future studies through comprehensive measurements and detailed characterization in organic composition changes.

[Figure]

Fig. R8 The fraction of ALWC contributed by organic aerosol and inorganic aerosol, respectively, during the winter of 2013-2014 (a) and the winter of 2018-2019 (b).

The methodology for SHAP analysis is not very clear as well. The validation of the output results from the machine learning model shall be displayed.

**Response:** We thank the referee's comment and suggestion. The RF model, in conjunction with SHAP analysis, was employed to assess the significance of variables in predicting SOA. The model's performance was validated based on the root mean square error (RMSE) and the correlation between predicted and observed values of the test data ($R^2$), as detailed in Table S2. Size ranges with $R^2 > 0.5$ in both 2013-2014 and 2018-2019 were exclusively chosen for comparison in our study.

According to the referee's suggestion, a more detailed explanation of the methodology for SHAP analysis has been added in section 2.5 in the revised manuscript as follows:

"……The model's performance was evaluated based on the root mean square error (RMSE) and the correlation between predicted and observed values of the test data ($R^2$) (Table S2 and Fig. S9). ……Moreover, the SHapley Additive explanation (SHAP) algorithm was applied to evaluate the importance of each variable in predicting SOA. SHAP is a novel interpretable method for machine learning models based on the concept of Sharpley value in game theory, which can help interpret the complex model predictions (nonlinear fitting) and quantify the

input variables' contributions to a single prediction (SOA) with physical significance (Lundberg et al., 2020). The functional expression of SHAP can be defined as follows:

$$f(x_i)=\phi_0(f,x)+\sum_{i=1}^{M}\phi_j(f,x_i) \qquad (3)$$

where $f(x_i)$ denotes the predicted value produced for each sample ($x_i$) with M features. $\phi_0(f,x)$ serves as the base value representing the anticipated output value of the RF model across the dataset. Additionally, $\phi_j(f,xi)$ represents the SHAP value delineating the influence of feature j in the sample ($x_i$) on the prediction of that specific sample (Lundberg et al., 2020)".

Meanwhile, the correlation between the predicted and observed values of the test data in the RF-SHAP analysis (shown in Fig. R9) has been added as Fig. S9 in the revised supplement accordingly.

[Figure]

Fig. R9 Correlation between the predicted and observed SOA fractions across different size ranges in the winters of 2013-2014 and 2018-2019, respectively.

The meaning of Figure 6 is not easy to follow for readers who are not farmilar with SHAP model, and more explanation is needed here. For the water in Figure 6b, it seems that the water contributes negative SOA formation when the water mass concentration is low and positive SOA formation when the water concentration is high. How to explain this? The importance of water to SOA formation seems to decrease from "225-317 nm" to "447-631 nm" and then increase when the particle size increases. What might cause this?

**Response:** We appreciate the referee's suggestion, and additional explanation for Fig. 6 has been incorporated into the revised manuscript accordingly. Within Fig. 6, the variables are

arranged in descending order of importance from top to bottom. A negative or positive SHAP value signifies that the variable contributes more, while a SHAP value closer to zero indicates less contribution to the prediction. As observed in Fig. 6b, aerosol water exhibited predominantly positive SHAP values across most data points, signifying a positive contribution to SOA formation in the majority of cases during the sampling period. Furthermore, the positive impact of aerosol water on SOA formation was particularly pronounced under high aerosol water conditions, with the SHAP value of aerosol water increasing as aerosol water levels intensified, implying that SOA formation may be predominantly influenced by aqueous-phase processes. Conversely, a small proportion of data points displayed negative SHAP values for aerosol water, indicating that periods with low aerosol water levels during the sampling period were less conducive to SOA formation, aligning with the dominant effects of aqueous-phase processes on SOA formation.

Furthermore, the variables in Fig. 6 are arranged in descending order of importance from top to bottom. Consequently, the significance of aerosol water in SOA formation increases progressively from "225-317 nm" to "447-631 nm" and then to "890-1256 nm". We speculate that the contrasting display order of variable importance between Fig. 5 (ordered in increasing importance from top to bottom) and Fig. 6 might lead to potential misunderstandings regarding their results and information. Therefore, in the revised manuscript, we have adjusted the variable importance ranking in Fig. 5 to be listed from top to bottom by importance, aligning it with Fig. 6, to ensure a clearer understanding of the results (as shown in Fig. R10).

In the revised manuscript lines 358-377, additional explanation regarding Fig. 6 has been added as follows:

"Similarly, ……The variables are arranged in descending order of importance from top to bottom. A negative or positive SHAP value signifies that the variable contributes more, while a SHAP value closer to zero indicates less contribution to the SOA formation…… Compared to winter 2013-2014, Ox showed less contribution to SOA formation with a close to zero SHAP value more frequently in winter 2018-2019. Conversely, aerosol water exhibited predominantly positive SHAP values across most data points, signifying a positive contribution to SOA formation in the majority of cases during the sampling period, particularly noticeable at larger particle sizes. Furthermore, the positive impact of aerosol water on SOA formation was particularly pronounced under high aerosol water conditions, with the SHAP value of aerosol water increasing as aerosol water levels intensified, implying that SOA formation may be predominantly influenced by aqueous-phase processes. These results suggest enhanced positive impact of aerosol water on SOA formation in winter 2018-2019. We suppose that ……".

[Figure]

Fig. R10 Random forest analysis for relative importance of factors impacting SOA formation during winter 2013-2014 (a-e) and winter 2018-2019 (f-j) in Xi'an.

How to explain the temperature effect on SOA formation in Fig. 6. More explanation is needed here since temperature is the most important factors which influence the SOA formation here.

**Response:** We thank the referee's suggestion. As shown in Fig. 6a, a positive SHAP value was observed more frequently for Ox, and the SHAP value of temperature increased positively as temperature increased across the size range of 225-1256 nm in 2013-2014, suggesting the dominant formation processes of SOA from photochemical oxidation in winter 2013-2014. These findings are consistent with the results presented in Elser et al., 2016. In comparison, as illustrated in Fig. 6b, apart from the significant contributions of aerosol water, temperature also exhibited notable impacts on SOA formation across the size range of 225-1256 nm in 2018-2019. The SHAP value of temperature increased with rising temperatures, indicating that higher temperatures also facilitate SOA formation in 2018-2019. This could be attributed to that the elevated temperatures may accelerate the chemical oxidation reactivity of precursors, thereby enhancing the processes involved in SOA formation (Jian et al., 2012; Chen et al., 2020). Similarly, a recent study indicated that the SHAP value of air temperature for $PM_{2.5,meteo}$ reached its peak around 10°C, suggesting that 10°C could be the most favorable temperature for winter haze pollution (Hou et al., 2022). This temperature aligns with the range observed (-12°C to 8°C) during winter 2018-2019 in our study.

In the revised manuscript lines 380-386, the discussion regarding the effects of temperature on SOA formation has been added, which now reads as follows:

"……and thus efficient aqueous-phase formation of SOA (Lv et al., 2023). Apart from the significant contributions of aerosol water, temperature also exhibited notable impacts on SOA formation across the size range of 225-1256 nm. The SHAP value of temperature increased with rising temperatures, indicating that higher temperatures also facilitate SOA formation in winter 2018-2019. This could be attributed to that the elevated temperatures may accelerate the

chemical oxidation reactivity of precursors, thereby enhancing the processes involved in SOA formation (Jian et al., 2012; Chen et al., 2020)……"

Minor comments

Line 121: How many factors are the signals with SNR with 0.2 <SNR<3 downweighed?

**Response:** We thank the referee's comment. The signals with $0.2 < S/N < 3$ were downweighed by a factor of 2 for both the 2013-2014 and 2018-2019 datasets. In the revised manuscript lines 139-140, the sentence has been updated, which now reads "……the ions with signal-to-noise $(S/N) < 0.2$ were removed and those with $0.2 < S/N < 3$ were down-weighted by a factor of 2 (Xu et al., 2021)".

Line 142: The formula is missing a sequence number.

**Response:** Change made. A sequence number of "(2)" has been added for the formula in the revised manuscript.

The color code of Fig. 2 is not very clear, especially for the small particle size (112-631nm), please modify it so that the readers can see it more clearly.

**Response:** We thank the referee's suggestion and change made. The color code of Fig. 2 has been updated in the revised manuscript, as shown in Fig. R11.

[Figure]

Fig. R11 Variations of the fractions of nitrate and sulfate, as well as the ratio of $NO_3^-/SO_4^{2-}$ at different NR-PM$_{2.5}$ levels in winter 2013-2014 (a, b, c) and winter 2018-2019 (d, e, f), respectively.

Line 90, Is eptof data used here or the regular PToF data? Please clarify.

**Response:** We thank the referee's suggestion and change made. Owing to the version disparity

between the HR-ToF-AMS and LToF-AMS employed in our study, regular PToF data was utilized in 2013-2014, while ePToF data was utilized in 2018-2019. In response to referee 1's comment above, a clarification and explanation of the differences between the two instruments have been added in the revised manuscript lines 92-107, which now read as follows:

"……The size-resolved NR-PM$_{2.5}$ composition was analyzed using a high resolution-time of flight-aerosol mass spectrometer (HR-ToF-AMS, Aerodyne Research Inc.) in PToF mode during the winter of 2013-2014, and a soot particle-long time of flight-aerosol mass spectrometer (SP-LToF-AMS, Aerodyne Research Inc.) in efficient PToF (ePToF) mode in the winter of 2018-2019, respectively. In our study, the SP-LToF-AMS operated as a standard LToF-AMS in "laser off" mode, exclusively measuring non-refractory aerosols, aligning with the HR-ToF-AMS measurements. The LToF-AMS is an upgraded version of the HR-ToF-AMS, employing the same quantification principle for non-refractory organic and inorganic components while featuring an extended ToF mass spectrometer chamber to enhance mass resolution and a multiplexed chopper in ePToF mode for a higher sizing duty cycle during particle sampling. To ensure data comparability, ionization efficiency (IE) calibration and size calibration utilizing NH$_4$NO$_3$ were conducted for both instruments during the 2013-2014 and 2018-2019 campaigns, respectively. Detailed information on instrument operation and calibration can also be found in previous studies (Elser et al., 2016; Duan et al., 2022)".

References:

Guo, H., Xu, L., Bougiatioti, A., Cerully, K.M., Capps, S.L., Hite, J.R., Carlton, A.G., Lee, S.H., Bergin, M.H., Ng, N.L., Nenes, A., Weber, R.J., 2015. Fine-particle water and pH in the southeastern United States. Atmospheric Chemistry and Physics 15, 5211-5228.

References:

Canagaratna, M. R., Jayne, J. T., Jiménez, J. L., Allan, J. D., Alfarra, M. R., Zhang, Q., Onasch, T. B., Drewnick, F., Coe, H., Middlebrook, A., Delia, A., Williams, L. R., Trimborn, A. M., Northway, M. J., DeCarlo, P. F., Kolb, C. E., Davidovits, P., and Worsnop, D. R.: Chemical and Microphysical Characterization of Ambient Aerosols with the Aerodyne Aerosol Mass Spectrometer, Mass Spectrom. Rev., 26, 185–222, 2007.

Cerully, K. M., Bougiatioti, A., Hite Jr., J. R., Guo, H., Xu, L., Ng, N. L., Weber, R., and Nenes, A.: On the link between hygroscopicity, volatility, and oxidation state of ambient and water-soluble aerosols in the southeastern United States, Atmos. Chem. Phys., 15, 8679–8694, https://doi.org/10.5194/acp-15-8679-2015, 2015.

Chen, G., Canonaco, F., Tobler, A., Aas, W., Alastuey, A., Allan, J., Atabakhsh, S., Aurela, M., Baltensperger, U., Bougiatioti, A., Brito, J. F. D., Ceburnis, D., Chazeau, B., Chebaicheb, H., Daellenbach, K. R., Ehn, M., Haddad, I. E., Eleftheriadis, K., Favez, O., Flentje, H.,

Font, A., Fossum, K., Freney, E., Gini, M., Green, D. C., Heikkinen, L., Herrmann, H., Kalogridis, A., Keernik, H., Lhotka, R., Lin, C., Lunder, C., Maasikmets, M., Manousakas, M. I., Marchand, N., Marin, C., Marmureanu, L., Mihalopoulos, N., Močnik, G., Nęcki, J., O'Dowd, C., Ovadnevaite, J., Peter, T., Petit, J., Pikridas, M., Platt, S. M., Pokorná, P., Poulain, L., Priestman, M., Riffault, V., Rinaldi, M., Różański, K., Schwarz, J., Sciare, J., Simon, L., Skiba, A., Slowik, J. G., Sosedova, Y., Stavroulas, I., Styszko, K., Teinemaa, E., Timonen, H., Tremper, A., Vasilescu, J., Via, M., Vodička, P., Wiedensohler, A., Zografou, O., Minguillón, M. C., Prévôt, A. S. H.: European aerosol phenomenology − 8: Harmonised source apportionment of organic aerosol using 22 Year-long ACSM/AMS datasets, Environ. Int., 166, 107325, https://doi.org/10.1016/j.envint.2022.107325, 2022.

Chen, Z., Chen, D., Zhao, C., Kwan, M.-p., Cai, J., Zhuang, Y., Zhao, B., Wang, X., Chen, B., Yang, J., Li, R., He, B., Gao, B., Wang, K., and Xu, B.: Influence of meteorological conditions on PM2.5 concentrations across China: A review of methodology and mechanism, Environ. Int., 139, 105558, https://doi.org/10.1016/j.envint.2020.105558, 2020.

Cubison, M. J., Ortega, A. M., Hayes, P. L., Farmer, D. K., Day, D., Lechner, M. J., Brune, W. H., Apel, E., Diskin, G. S., Fisher, J. A., Fuelberg, H. E., Hecobian, A., Knapp, D. J., Mikoviny, T., Riemer, D., Sachse, G. W., Sessions, W., Weber, R. J., Weinheimer, A. J., Wisthaler, A., and Jimenez, J. L.: Effects of aging on organic aerosol from open biomass burning smoke in aircraft and laboratory studies, Atmos. Chem. Phys., 11, 12049–12064, https://doi.org/10.5194/acp-11-12049-2011, 2011.

Duan, J., Huang, R. J., Gu, Y., Lin, C., Zhong, H., Xu, W., Liu, Q., You, Y., Ovadnevaite, J., Ceburnis, D., Hoffmann, T., and O'Dowd, C.: Measurement report: Large contribution of biomass burning and aqueous-phase processes to the wintertime secondary organic aerosol formation in Xi'an, Northwest China, Atmos. Chem. Phys., 22, 10139–10153, https://doi.org/10.5194/acp-22-10139-2022, 2022.

Elser, M., Huang, R. J., Wolf, R., Slowik, J. G., Wang, Q.,Canonaco, F., Li, G., Bozzetti, C., Daellenbach, K. R., Huang, Y., Zhang, R., Li, Z., Cao, J., Baltensperger, U., El-Haddad, I., and André, P.: New insights into PM2.5 chemical composition and sources in two major cities in China during extreme haze events using aerosol mass spectrometry, Atmos. Chem. Phys., 16, 3207–3225, https://doi.org/10.5194/acp-16-3207-2016, 2016.

Gunthe, S. S., Rose, D., Su, H., Garland, R. M., Achtert, P., Nowak, A., Wiedensohler, A., Kuwata, M., Takegawa, N., Kondo, Y., Hu, M., Shao, M., Zhu, T., Andreae, M. O., and Pöschl, U.: Cloud condensation nuclei (CCN) from fresh and aged air pollution in the megacity region of Beijing, Atmos. Chem. Phys., 11, 11023–11039, https://doi.org/10.5194/acp-11-11023-2011, 2011.

Guo, H., Xu, L., Bougiatioti, A., Cerully, K. M., Capps, S. L., Hite Jr., J. R., Carlton, A. G., Lee, S.-H., Bergin, M. H., Ng, N. L., Nenes, A., and Weber, R. J.: Fine-particle water and pH in the southeastern United States, Atmos. Chem. Phys., 15, 5211–5228,

https://doi.org/10.5194/acp-15-5211-2015, 2015.

Hennigan, C. J., Izumi, J., Sullivan, A. P., Weber, R. J., and Nenes, A.: A critical evaluation of proxy methods used to estimate the acidity of atmospheric particles, Atmos. Chem. Phys., 15, 2775–2790, https://doi.org/10.5194/acp-15-2775-2015, 2015.

Hou, L. L., Dai, Q. L., Song, C. B., Liu, B., Guo, F. Z., Dai, T. J., Li, L. X., Liu, B. S., Bi, X. H., Zhang, Y. F., and Feng, Y. C.: Revealing drivers of haze pollution by explainable machine learning, Environ. Sci. Tech. Let., 9, 112–119, https://doi.org/10.1021/acs.estlett.1c00865, 2022.

Hu, W. W., Hu, M., Hu, W., Jimenez, J. L., Yuan, B., Chen, W., Wang, M., Wu, Y., Chen, C., Wang, Z., Peng, J., Zeng, L., and Shao, M.: Chemical composition, sources, and aging process of submicron aerosols in Beijing: Contrast between summer and winter, J. Geophys. Res.-Atmos., 121, 1955–1977, https://doi.org/10.1002/2015JD024020, 2016.

Huang, R.-J., He, Y., Duan, J., Li, Y., Chen, Q., Zheng, Y., Chen, Y., Hu, W., Lin, C., Ni, H., Dai, W., Cao, J., Wu, Y., Zhang, R., Xu, W., Ovadnevaite, J., Ceburnis, D., Hoffmann, T., and O'Dowd, C. D.: Contrasting sources and processes of particulate species in haze days with low and high relative humidity in wintertime Beijing, Atmos. Chem. Phys., 20, 9101–9114, https://doi.org/10.5194/acp-20-9101-2020, 2020.

Jian, L., Zhao, Y., Zhu, Y. P., Zhang, M. B., and Bertolatti, D.: An application of arima model to predict submicron particle concentrations from meteorological factors at a busy roadside in hangzhou, China, Sci. Total Environ., 426, 336–345, 2012.

Jimenez, J. L., Jayne, J. T., Shi, Q., Kolb, C. E., Worsnop, D. R., Yourshaw, I., Seinfeld, J. H., Flagan, R. C., Zhang, X., Smith, K. A., Morris, J. W., and Davidovits, P.: Ambient aerosol sampling with an Aerosol Mass Spectrometer, J. Geophys. Res.-Atmos., 108, 8425, doi:10:1029/2001JD001213, 2003.

Lundberg, S. M., Erion, G., Chen, H., DeGrave, A., Prutkin, J. M., Nair, B., Katz, R., Himmelfarb, J., Bansal, N., and Lee, S.-I.: From local explanations to global understanding with explainable AI for trees, Nat. Mach. Intell., 2, 56–67, https://doi.org/10.1038/s42256-019-0138-9, 2020.

Lv, S. J., Wu, C., Wang, F. L., Liu, X. D., Zhang, S., Chen, Y. B., Zhang, F., Yang, Y., Wang, H. L., Huang, C., Fu, Q. Y., Duan, Y. S., and Wang, G. H.: Nitrate-enhanced gas-to-particle-phase partitioning of water-soluble organic compounds in Chinese urban atmosphere: implications for secondary organic aerosol formation, Environ. Sci. Technol. Lett., 10, 14–20, https://doi.org/10.1021/acs.estlett.2c00894, 2023.

Ng, N. L., Canagaratna, M. R., Zhang, Q., Jimenez, J. L., Tian, J., Ulbrich, I. M., Kroll, J. H., Docherty, K. S., Chhabra, P. S., Bahreini, R., Murphy, S. M., Seinfeld, J. H., Hildebrandt, L., Donahue, N. M., DeCarlo, P. F., Lanz, V. A., Prévôt, A. S. H., Dinar, E., Rudich, Y., and Worsnop, D. R.: Organic aerosol components observed in Northern Hemispheric datasets from Aerosol Mass Spectrometry, Atmos. Chem. Phys., 10, 4625–4641, https://doi.org/10.5194/acp-10-4625-2010, 2010.

Ng, N. L., Canagaratna, M. R., Jimenez, J. L., Chhabra, P. S., Seinfeld, J. H., and Worsnop, D. R.: Changes in organic aerosol composition with aging inferred from aerosol mass spectra, Atmos. Chem. Phys., 11, 6465–6474, https://doi.org/10.5194/acp-11-6465-2011, 2011.

Song, S., Gao, M., Xu, W., Shao, J., Shi, G., Wang, S., Wang, Y., Sun, Y., and McElroy, M. B.: Fine-particle pH for Beijing winter haze as inferred from different thermodynamic equilibrium models, Atmos. Chem. Phys., 18, 7423–7438, https://doi.org/10.5194/acp-18-7423-2018, 2018.

Sun, Y. L., Du, W., Fu, P., Wang, Q., Li, J., Ge, X., Zhang, Q., Zhu, C., Ren, L., Xu, W., Zhao, J., Han, T., Worsnop, D. R., and Wang, Z.: Primary and secondary aerosols in Beijing in winter: sources, variations and processes, Atmos. Chem. Phys., 16, 8309–8329, https://doi.org/10.5194/acp-16-8309-2016, 2016.

Sun, Y., Xu, W., Zhang, Q., Jiang, Q., Canonaco, F., Prévôt, A. S. H., Fu, P., Li, J., Jayne, J., Worsnop, D. R., and Wang, Z.: Source apportionment of organic aerosol from 2-year highly time-resolved measurements by an aerosol chemical speciation monitor in Beijing, China, Atmos. Chem. Phys., 18, 8469–8489, https://doi.org/10.5194/acp-18-8469-2018, 2018.

Tilgner, A., Schaefer, T., Alexander, B., Barth, M., Collett Jr., J. L., Fahey, K. M., Nenes, A., Pye, H. O. T., Herrmann, H., and McNeill, V. F.: Acidity and the multiphase chemistry of atmospheric aqueous particles and clouds, Atmos. Chem. Phys., 21, 13483–13536, https://doi.org/10.5194/acp-21-13483-2021, 2021.

Wang, G., Zhang, R., Gomez, M. E., Yang, L., Levy Zamora, M., Hu, M., Lin, Y., Peng, J., Guo, S., Meng, J., Li, J., Cheng, C., Hu, T., Ren, Y., Wang, Y., Gao, J., Cao, J., An, Z., Zhou, W., Li, G., Wang, J., Tian, P., Marrero-Ortiz, W., Secrest, J., Du, Z., Zheng, J., Shang, D., Zeng, L., Shao, M., Wang, W., Huang, Y., Wang, Y., Zhu, Y., Li, Y., Hu, J., Pan, B., Cai, L., Cheng, Y., Ji, Y., Zhang, F., Rosenfeld, D., Liss, P. S., Duce, R. A., Kolb, C. E., and Molina, M. J.: Persistent sulfate formation from London Fog to Chinese haze, P. Natl. Acad. Sci. USA, 113, 13630, https://doi.org/10.1073/pnas.1616540113, 2016.

Wu, Z. J., Zheng, J., Shang, D. J., Du, Z. F., Wu, Y. S., Zeng, L. M., Wiedensohler, A., and Hu, M.: Particle hygroscopicity and its link to chemical composition in the urban atmosphere of Beijing, China, during summertime, Atmos. Chem. Phys., 16, 1123–1138, https://doi.org/10.5194/acp-16-1123-2016, 2016.

Xie, Y., Wang, G., Wang, X., Chen, J., Chen, Y., Tang, G., Wang, L., Ge, S., Xue, G., Wang, Y., and Gao, J.: Nitrate-dominated $PM_{2.5}$ and elevation of particle pH observed in urban Beijing during the winter of 2017, Atmos. Chem. Phys., 20, 5019–5033, https://doi.org/10.5194/acp-20-5019-2020, 2020.

Xu, J., Shi, J., Zhang, Q., Ge, X., Canonaco, F., Prévôt, A. S. H., Vonwiller, M., Szidat, S., Ge, J., Ma, J., An, Y., Kang, S., and Qin, D.: Wintertime organic and inorganic aerosols in Lanzhou, China: sources, processes, and comparison with the results during summer, Atmos. Chem. Phys., 16, 14937–14957, https://doi.org/10.5194/acp-16-14937-2016, 2016.

Xu, J., Ge, X., Zhang, X., Zhao, W., Zhang, R., and Zhang, Y.: COVID-19 impact on the concentration and composition of submicron particulate matter in a typical city of Northwest China, Geophys. Res. Lett., 47, e2020GL089035, https://doi.org/10.1029/2020GL089035, 2020.

Xu, L., Suresh, S., Guo, H., Weber, R. J., and Ng, N. L.: Aerosol characterization over the southeastern United States using high-resolution aerosol mass spectrometry: spatial and seasonal variation of aerosol composition and sources with a focus on organic nitrates, Atmos. Chem. Phys., 15, 7307–7336, https://doi.org/10.5194/acp-15-7307-2015, 2015.

Ye, Z., Liu, J., Gu, A., Feng, F., Liu, Y., Bi, C., Xu, J., Li, L., Chen, H., Chen, Y., Dai, L., Zhou, Q., and Ge, X.: Chemical characterization of fine particulate matter in Changzhou, China, and source apportionment with offline aerosol mass spectrometry, Atmos. Chem. Phys., 17, 2573–2592, https://doi.org/10.5194/acp-17-2573-2017, 2017.

Zhang, J. K., Sun, Y., Liu, Z. R., Ji, D. S., Hu, B., Liu, Q., and Wang, Y. S.: Characterization of submicron aerosols during a month of serious pollution in Beijing, 2013, Atmos. Chem. Phys., 14, 2887–2903, https://doi.org/10.5194/acp-14-2887-2014, 2014.

Zheng, Y., Chen, Q., Cheng, X., Mohr, C., Cai, J., Huang, W., Shrivastava, M., Ye, P., Fu, P., Shi, X., Ge, Y., Liao, K., Miao, R., Qiu, X., Koenig, T. K., and Chen, S.: Precursors and Pathways Leading to Enhanced Secondary Organic Aerosol Formation during Severe Haze Episodes, Environ. Sci. Technol., 55, 15680–15693, https://doi.org/10.1021/acs.est.1c04255, 2021.

Zheng, Y., Miao, R. Q., Zhang, Q., Li, Y. W., Cheng, X., Liao, K. R., Koenig, T. K., Ge, Y. L., Tang, L. Z., Shang, D. J., Hu, M., Chen, S. Y., and Chen, Q.: Secondary Formation of Submicron and Supermicron Organic and Inorganic Aerosols in a Highly Polluted Urban Area, J. Geophys. Res.-Atmos., 128, e2022JD037865, https://doi.org/10.1029/2022jd037865, 2023.